# Effects of different exercise training programs on the functional performance in fibrosing interstitial lung diseases: A randomized trial

**Hatem Essam[1], Nashwa Hassan Abdel Wahab[1], Gihan Younis[2], Enas El-sayed[1], Hanaa Shafiek[1] ***

**1** Department of Chest diseases, Faculty of Medicine, Alexandria University, Alexandria, Egypt,
**2** Department of Physical Medicine, Rheumatology and Rehabilitation, Faculty of medicine, Alexandria University, Alexandria, Egypt

* whitecoat.med@gmail.com

## Abstract

### Objectives

We aimed to compare the effects of different aerobic exercise training (ET) programs on respiratory performance, exercise capacity, and quality of life in fibrosing interstitial lung diseases (f-ILD).

### Methods

A case-control study where 31 patients with f-ILD diagnosis based on chest high-resolution computed tomography were recruited from Main Alexandria University hospital-Egypt. Ten patients were randomly assigned for only lower limbs (LL) endurance training program, and 10 patients for upper limbs, lower limbs, and breathing exercises (ULB) program for consecutive 18 sessions (3 sessions/week for 6 consecutive weeks). Eleven patients who refused to participate in the ET program were considered as control. All patients were subjected for St George's respiratory questionnaire (SGRQ), 6-minute walk test (6-MWT), forced spirometry and cardiopulmonary exercise testing (CPET) before and after ET programs.

### Results

Fibrosing non-specific interstitial pneumonia (NSIP) and collagenic associated-ILD were the commonest pathologies among the ET groups (30% each) with mean age of $44.4\pm12.25$ and $41.90\pm7.58$ years for LL and ULB groups respectively and moderate-to-severe lung restriction. 6-MWT and SGRQ significantly improved after both ET programs ($p<0.001$). Peak oxygen consumption ($VO_2$) improved significantly after both LL training (median of 22 (interquartile range (IQR) = 17.0–24.0) vs. 17.5 (IQR = 13.0–23.0) ml/kg/min, p = 0.032) and ULB training (median of 13.5 (IQR = 11.0–21.0) vs. 10.5 (IQR = 5.0–16.0) ml/kg/min, p = 0.018). Further, maximal work load and minute ventilation (VE) significantly improved after both types of ET training ($p<0.05$); however, neither ventilation equivalent ($VE/VCO_2$) nor FVC% improved after ET (p = 0.052 and 0.259 respectively). There were no statistically

**Data Availability Statement:** All relevant data are within the manuscript and its Supporting Information files. The raw excel file is also now

available on the public repository (Fig share). The dataset is currently as a private link that will be available as public on acceptance. URL https://figshare.com/s/fd724ded44c70a86cef5 doi: 10.6084/m9.figshare.19640214.

**Funding:** The author(s) received no specific funding for this work.

**Competing interests:** The authors have declared that no competing interests exist.

significant important differences between LL and ULB training programs regarding 6-MWT, SGRQ or CPET parameters ($p>0.05$).

## Conclusions

ET was associated with improvements in exercise capacity and quality of life in f-ILD patients irrespective of the type of ET program provided.

## Introduction

Interstitial lung diseases (ILDs) are a heterogeneous group of diseases characterized by pulmonary parenchymal inflammation and fibrosis [1]. Various subtypes of ILDs are referred to as fibrotic ILDs (f-ILDs) with overlapping in the clinical features, since they have an insidious onset of dry cough, shortness of breath, especially progressive exertion, and bibasilar crackles. They also share morphological characteristics and typical pathological mechanisms, as they are distinguished by the existence of diffuse and permanent fibrous lesions of the lung interstitium and alveolar parenchyma leading to the concept of a progressive fibrosing phenotype that can be applied to a variety of f-ILDs [1].

Idiopathic pulmonary fibrosis (IPF) is the most common subtype of f-ILDs [2]. However, other ILD subtypes also have a progressive fibrosing phenotype. These include fibrotic hypersensitivity pneumonitis (HP), unclassifiable ILD, non-specific interstitial pneumonia (NSIP), connective tissue diseases associated ILDs, organizing pneumonia, ILD associated with occupational exposures and rarely sarcoidosis. Progressive f-ILDs are associated with high mortality [3–5]. Decline in lung function and worsening of symptoms are reflecting the cardinal features of progressive f-ILDs which results in exercise limitation and marked deterioration of health-related quality of life (HRQoL) [6–8].

Exercise limitation in ILDs is multifactorial, with contributions of impairment of gas exchange and pulmonary circulation [9], ventilatory limitation [10], and peripheral muscle dysfunction [11, 12]. Pulmonary rehabilitation (PR) is gaining wide acceptance in the management of chronic respiratory diseases especially chronic obstructive pulmonary disease (COPD) in the last years [13]. Exercise training (ET) is an integral component of PR for ILDs, including resistance and endurance training which is important in increasing cardiorespiratory fitness and exercise capacity [14]. Despite weak recommendation for PR in the guidelines of management of IPF [15], PR has shown benefits in patients with ILDs irrespective of the underlying pathology in terms of reduce the severity of symptoms, improvement of functional exercise capacity and HRQoL [16].

We hypothesized that patients with f-ILDs could get benefits from various types of ET. We aimed to compare the effects of different aerobic ET programs (namely lower limbs only (LL) versus upper limbs, lower limbs, and breathing exercises (ULB)) on respiratory performance, exercise capacity, and HRQoL in f-ILDs patients.

## Materials and methods

### Study design and ethics

A prospective randomized case-control study with short-term follow-up that enrolled patients with diagnosis of f-ILDs based on chest high-resolution computed tomography (HRCT). The study was conducted at Alexandria Main University hospitals, Alexandria, Egypt between

January 2020 and January 2021. The study was approved by local ethical committee of Alexandria Faculty of Medicine of Egypt (protocol ID: 0201313). The protocol was registered in ClinicalTrials.gov (ID number: NCT05227443). All the participants signed an informed written consent.

## Patients' characteristics

Adult patients aged more than 18 years–with age range between 25 to 70 years–who were previously diagnosed as f-ILDs based on HRCT radiological features, in addition to the restrictive or mixed pattern in spirometric results were enrolled. Patients with motor disabilities, cardiovascular diseases (as acute heart failure, unstable angina or recent myocardial infarction), cognitive impairments, history of cerebrovascular accident, active cancer, and a life expectancy below 3 months were excluded from the study. Thirty-one patients with f-ILDs were enrolled, of whom 11 patients refused to participate in ET programs or unable to participate due to morbid obesity or living outside the influence of the hospital or voluntary withdrew from the study were considered as control group (Fig 1) whom met the same inclusion and exclusion criteria of the study. Accordingly, 20 patients were randomized equally for either LL only aerobic ET or ULB and further reanalyzed.

All patients were subjected to complete history including modified Medical Research Council (mMRC) dyspnea scale [17] (with detailed description in S1 File) and smoking history; complete physical examination including anthropometric measures and body mass index (BMI); and HRCT of chest. A detailed drug history was also taken from all participants including the use of oral corticosteroids, immunosuppressive medication and antifibrotic treatment; and we did not modify either the dose or pharmacological drug was taken by any patient

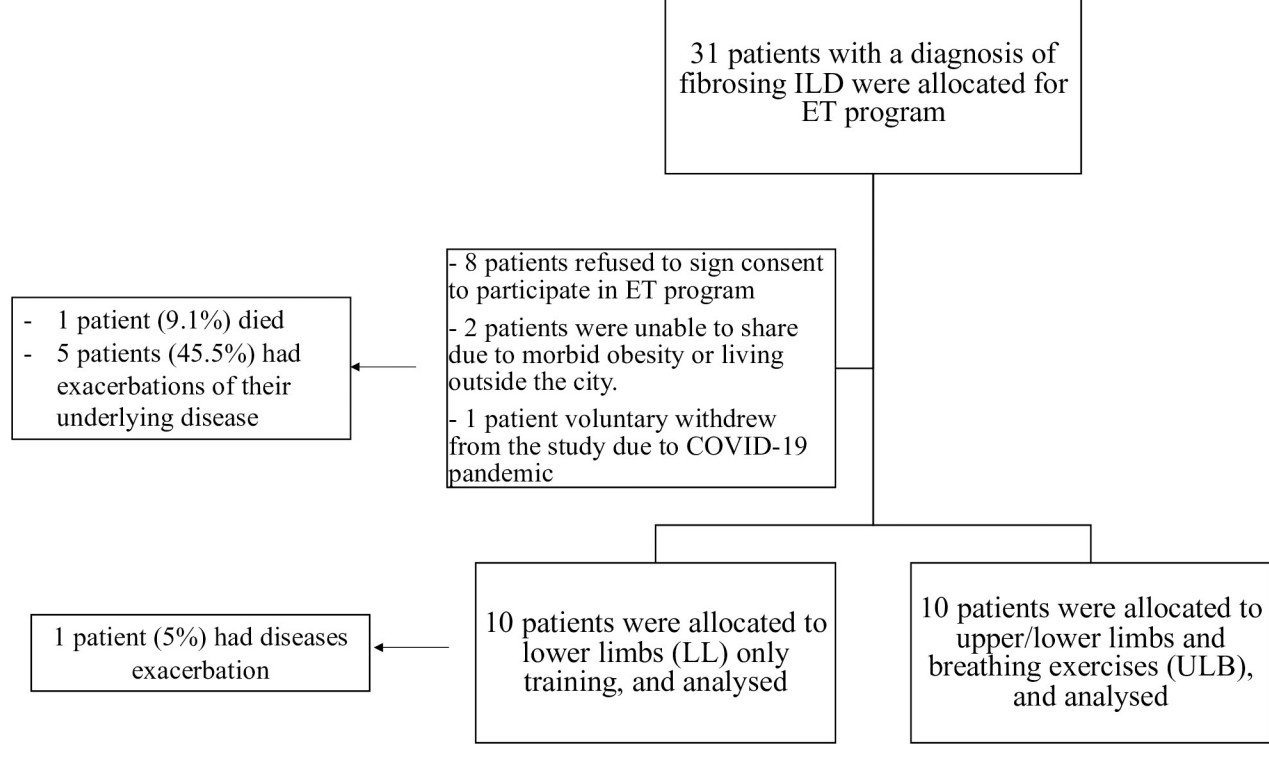

**Fig 1. Flow chart of the studied population.**

throughout the study duration to avoid any risk bias in our results. Forced spirometry and 6-minute walk test (6MWT) [18] according to international guidelines were performed for all patients before ET. St. George's Respiratory Questionnaire (SGRQ) [19] for assessment of HRQoL was obtained also before ET which includes 3 categories: symptoms component (frequency and severity), activity component and impact component (social functioning, psychological disturbances). mMRC dyspnea scale, forced spirometry, 6MWT, and SGRQ were repeated by the end of ET program. Regarding the control group, they were followed up by phone calls for symptomatology change and exacerbations history due to the COVID-19 pandemic state.

## Cardiopulmonary exercise test (CPET)

CPET was performed for all the patients before and after the sessions of ET. CPET was performed using the Ergocard clinical exercise testing system (Ergocard Clinical, Medisoft, Sorinnes, Belgium) according to American Thoracic Society (ATS) guidelines [20]. The reference values of Jones et al [21] were considered. Incremental CPET protocol was conducted on an electronically braked cycle ergometer using ExpAir software (ExpAir, version 1.34, Medisoft, Sorinnes, Belgium). All CPET parameters were recorded at baseline (i.e., at zero watts during the warming phase of CPET just before starting the incremental increase of work load) and at maximum work load (WL) achieved by the participants. These parameters included: work load (WL), minute ventilation (VE), oxygen consumption ($VO_2$), oxygen consumption / kilogram ($VO_2$/kg; which is considered as the peak $VO_2$ at the maximal WL), carbon dioxide output ($VCO_2$), heart rate (HR), respiratory rate (RR), oxygen pulse ($VO_2$/HR), dead space (VD), tidal volume (VT), systolic blood pressure (SBP), diastolic blood pressure (DBP), breathing reserve (BR), respiratory quotient (RQ), end-tidal carbon dioxide pressure ($P_{ET}CO_2$), end-tidal oxygen pressure ($P_{ET}O_2$), and oxygen saturation ($SpO_2$). The VE/$VCO_2$ was calculated.

## Exercise training (ET)

Twenty patients enrolled in ET programs: 10 patients performed LL only training and 10 patients performed ULB training. ET protocols were consistent with those recommended for use in PR programs for people with COPD [22]. The program continued for 6 weeks where the patients had 3 supervised sessions/week (a total of 18 sessions). Lower limb aerobic training was performed on a treadmill (S1 Fig in S1 File). The standards of exercise prescription were applied as previously described for chronic lung diseases [22, 23]. The program was individualized; as the initial duration, the initial intensity, and the rate of progression varied among patients based on their exercise tolerance.

The exercise intensity was measured as the percentage of the maximum heart rate determined from the equation (220 –age of the participant) [24]. During the 1st week, the patients exercised initially at low intensity exercise i.e., 50–60% of their maximum heart rate and short duration of usually 10 minutes that was broken into shorter intervals if needed (as cycles of 3 minutes training followed by 1–2 minutes of rest period). The 2nd week, an attempt was made to increase the performed work during training by increasing the duration of session by 5 minutes every 2 sessions with decreasing the intervals between training, and increasing the workload by 5% every 2–3 sessions according to patient's tolerance. The 3rd week, most patients were able to continue 30 minutes of aerobic exercise as 2 cycles of continuous 15 minutes aerobic training separated by one interval of rest at moderate exercise intensity of 64–76% of their maximum heart rate. The 4th– 6th week, the ET continued the achievement of 3rd week whereas most patients were able to exercise 30 minutes continuously at moderate exercise intensity which was the main target to achieve.

Upper limb exercise was performed on a wheel (S2 Fig in S1 File). Each session lasted for 15 minutes of continuous exercise which was well tolerated by patients (workload is less demanding). For breathing exercises, an incentive spirometry was used [25].

During ET session, $SpO_2$ and HR were measured regularly to ensure safety. Those on long term oxygen therapy (LTOT) performed the ET while continuously used their level of L/min. The patients who desaturated below predetermined cutoff values (often $SpO_2 < 90\%$) and known to be on with acceptable $SpO_2$ on room air, supplemental oxygen was used to exercise safely. In patients who kept on desaturating despite adequate oxygen support, the exercise session was divided into multiple short bouts in order to allow $SpO_2$ to recover and stay in a safe range [26].

## Outcomes

The changes of CPET parameters, mMRC dyspnea scale, 6-minute walk distance (6MWD), SGRQ, and forced spirometry were recorded as primary outcomes. Further, mortality and disease exacerbation during the follow up time were recorded as secondary outcomes.

## Statistical analysis

All the data were expressed as median and interquartile range (IQR) for the non-normal distribution of continuous data or mean ± standard deviation (SD) for the normal distribution of continuous data. Frequencies and percentages (%) were used to report categorical data. Chi-square test, one-way ANOVA test and Kruskal-Wallis test were used in the comparison between 3 groups as appropriate; while Student independent *t*-test, Paired *t*-test, Mann-Whitney test, Wilcoxon signed rank test were used as appropriate when comparing between 2 groups. Further, a multivariate logistic regression in relation to outcome was conducted using forward method after adjustment to the significant baseline covariates found between the control group and ET intervention groups. Odd ratio (OR) and confidence interval (CI) 95% was shown. A two-tailed *p*-value $< 0.05$ was considered statistically significant. SPSS package (Version 22.0. Armonk, NY: IBM Corp) was used for all analyses.

## Results

### Patients' characteristics

Table 1 shows the demographics and baseline clinical characteristics of all the participants. The patients involved in ET program appeared to be significantly younger in age (*p* = 0.004, Table 1), more males in LL program and control rather than ULB program (*p* = 0.048, Table 1), and mostly non-smokers (*p* = 0.035, Table 1). There was no statistically significant difference between groups regarding presence of comorbidities, associated pulmonary hypertension, duration of illness, mMRC dyspnea scale, baseline 6-minute walk distance (6MWD), baseline SGRQ and baseline spirometric parameters (*p* > 0.05, Table 1). The mean FVC was 49.0 ± 12.30% predicted in LL group, and 49.9 ± 11.89% predicted in ULB group indicated a moderate to severe lung restriction.

Fibrosing NSIP and collagenic associated-ILD were the commonest pathologies followed by chronic HP among ET groups while IPF and chronic HP were the commonest pathologies among control group without statistically significant difference (*p* = 0.138, Table 1). Corticosteroids was the commonest prescribed medication among all groups whereas six patients (54.5%) in the control group and 18 patients (90%) in the ET groups were treated with corticosteroids (*p* = 0.049, Table 1). Eight patients (40% of both ET groups) and 4 patients (36.4%) of control group were on LTOT (*p* = 0.285).

**Table 1. Baseline characteristics of the studied groups.**

| Character | Control group (n = 11) | LL group (n = 10) | ULB group (n = 10) | Sig. (p value) |
|---|---|---|---|---|
| Age (yrs); mean (±SD) | 57.91 ± 11.74 | 44.40 ± 12.25 | 41.90 ± 7.58 | 0.004* |
| Gender; (n, %) Male / Female | 7 (63.6) / 4 (36.4) | 6 (60) / 4 (40) | 2 (20) / 8 (80) | 0.048* |
| BMI (kg/m$^2$); mean (±SD) | 28.24 ± 8.10 | 25.33 ± 5.36 | 27.89 ± 6.92 | 0.590 |
| Smoking status; (n, %) Active smoker/ non-smoker | 4 (36.4) / 7 (63.6) | 2 (20) / 8 (80) | 0 (0) / 10 (100) | 0.035* |
| Smoking index (pk/yr); median (IQR) | 72.5 (57.5–80.0) | 80 (50.0–110.0) | 0 (0) | 0.625 |
| Duration of the disease (months) | 19.0 ± 13.8 | 24 ± 9.80 | 17.3 ± 15.39 | 0.508 |
| Comorbidities (Y); (n, %) | 5 (45.5) | 3 (30) | 4 (40) | 0.784 |
| Hypertension | 1 (9.1) | 1 (10) | 1 (10) | 0.918 |
| DM | 2 (18.2) | 2 (20) | 2 (20) | |
| Both | 2 (18.2) | 0 (0) | 1 (10) | |
| Associated PHT | 7 (63.6) | 6 (60) | 8 (80) | 0.433 |
| mMRC dyspnea scale; mean (±SD) | NA | 2.90 ± 0.74 | 3 ± 0.67 | 0.754 |
| SGRQ (total) | 64.3 ± 17.4 | 74.6 ± 16.73 | 75.8 ± 18.78 | 0.272 |
| 6MWD (meter) | 243.82 ± 138.99 | 276.0 ± 130.06 | 264.0 ± 95.13 | 0.834 |
| Diagnosis; (n, %) | | | | |
| IPF | 4 (36.4) | 2 (20) | 0 (0) | 0.166 |
| Chronic HP | 5 (45.5) | 1 (10) | 4 (40) | |
| Fibrotic NSIP | 2 (18.2) | 3 (30) | 3 (30) | |
| Collagenic ILD | 0 (0) | 3 (30) | 3 (30) | |
| Chronic Sarcoidosis | 0 (0) | 1 (10) | 0 (0) | |
| Spirometry; mean (±SD) | | | | |
| FVC (L) | 1.46 ± 0.57 | 1.97 ± 0.60 | 1.93 ± 0.53 | 0.140 |
| FVC (% predicted) | 44.4 ± 15.13 | 49.0 ± 12.30 | 49.9 ± 11.89 | 0.592 |
| FEV$_1$ (L) | 2.0 ± 0.95 | 1.99 ± 0.58 | 1.694 ± 0.38 | 0.523 |
| FEV$_1$ (% predicted) | 51.69 ± 28.04 | 55.7 ± 14.69 | 55.2 ± 13.56 | 0.894 |
| FEV$_1$ / FVC | 93.9 ± 6.9 | 91.1 ± 9.15 | 88.1 ± 8.43 | 0.374 |
| Baseline medications; (n, %) | | | | |
| Corticosteroids | 6 (54.5) | 9 (90) | 9 (90) | 0.049* |
| Pirfenidone | 1 (9.1) | 0 (0) | 0 (0) | 0.232 |
| Immunosuppressive steroid sparing | 0 (0) | 3 (30) | 2 (20) | 0.199 |
| Acetyl cysteine | 5 (45.5) | 5 (50) | 5 (50) | 0.833 |
| LTOT | 4 (36.4) | 2 (20) | 6 (60) | 0.285 |

Abbreviations; yrs: years, pk/yr: pack/ year, Y: yes; SD: standard deviation, n: number, NA: not assessed, IQR: interquartile range, BMI: body mass index, DM: diabetes mellitus, IPF: idiopathic pulmonary fibrosis, HP: hypersensitivity pneumonitis, ILD: interstitial lung disease, NSIP: non-specific idiopathic pneumonitis, PHT: pulmonary hypertension, FVC: forced vital capacity, FEV$_1$: forced expiratory volume in 1 second, LTOT: long term oxygen therapy, 6MWD: 6-minute walk distance.

* Significant p value < 0.05.

## CPET

The various parameters of CPET both at baseline (at 0 watts of WL) and maximal exercise before starting ET programs are shown in S1 Table in S1 File and Table 2 respectively. The $VO_2$, $VO_2$% predicted, $VO_2$/kg and oxygen pulse ($VO_2$/HR) at baseline (at 0 watts of WL) were significantly lower among ULB groups when compared to LL group and control group ($p < 0.05$, S1 Table in S1 File). At maximal WL during CPET, $SpO_2$ of the control group was significantly lower when compared to LL and ULB training groups ($p = 0.025$, Table 2) while

**Table 2. CPET variables at maximal exercise workload before ET among the studied groups.**

| Variable | Control group (n = 11) | LL group (n = 10) | ULB group (n = 10) | Sig. (p value) |
|---|---|---|---|---|
| Time (min) | 6.63 ± 1.16 | 6.61 ± 1.30 | 7.61 ± 1.61 | 0.203 |
| Work load (watts) | 48.22 ± 23.24 | 49.20 ± 14.48 | 37.88 ± 21.62 | 0.441 |
| Work load% | 35.2 ± 16.59 | 33.7 ± 14.94 | 30.4 ± 19.32 | 0.837 |
| VE (L/min) | 44.50 ± 18.41 | 44.18 ± 11.50 | 31.81 ± 7.78 | 0.067 |
| VE% | 40.6 ± 19.62 | 42.1 ± 14.87 | 33.0 ± 9.38 | 0.353 |
| Breathing reserve (%) | 46.56 ± 17.56 | 46.5 ± 16.59 | 58.30 ± 16.99 | 0.228 |
| VD/VT ratio | 0.27 (0.20–0.30) | 0.22 (0.20–0.25) | 0.22 (0.21–0.24) | 0.258 |
| $VO_2$ (L/min) | 1.17 (0.64–1.38) | 1.19 (0.94–1.39) | 0.79 (0.56–1.10) | 0.123 |
| $VO_2$% | 69 (29–88) | 63.0 (46.0–72.0) | 36.5 (29.0–46.0) | 0.090 |
| $VO_2$/kg (ml/kg/min) | 19 (10–20) | 17.5 (13.0–23.0) | 10.5 (5.0–16.0) | 0.160 |
| $VCO_2$ (L/min) | 0.69 (0.47–1.08) | 0.93 (0.64–1.03) | 0.63 (0.36–0.68) | 0.064 |
| Respiratory rate (br/min) | 47.21 ± 14.46 | 47.50 ± 7.89 | 40.0 ± 9.38 | 0.236 |
| RER | 0.64 ± 0.21 | 0.79 ± 0.12 | 0.70 ± 0.32 | 0.366 |
| HR (b/min) | 119 (104–120) | 140 (122.25–144) | 122 (103.5–139.5) | 0.288 |
| HR% | 77 (65.0–80.0) | 78 (69.2–84.8) | 69 (65.0–82.3) | 0.708 |
| $VE/VCO_2$ | 43.19 ± 10.66 | 40.85 ± 14.22 | 45.83 ± 10.96 | 0.675 |
| $VO_2$/HR (ml/beat) | 10.2 (4.10–10.40) | 8.7 (6.85–10.0) | 3.9 (3.80–5.65)[$] | 0.044* |
| $VO_2$/HR% | 71.2 (47.9–71.5) | 70.4 (0.56–0.85) | 47.9 (0.45–0.64) | 0.133 |
| $PETCO_2$ (mmHg) | 23.67 ± 6.16 | 25.9 ± 4.77 | 26.6 ± 6.15 | 0.520 |
| $PETO_2$ (mmHg) | 118.44 ± 5.32 | 114.5 ± 8.96 | 117.5 ± 9.64 | 0.557 |
| $SpO_2$ (%) | 82.11 ± 1.97[$] | 86.8 ± 5.25 | 87.8 ± 5.2 | 0.025* |

Abbreviations; VE: minute ventilation, br/min: breath/minute, $VO_2$: oxygen consumption, $VCO_2$: carbon dioxide output, HR: heart rate, $VO_2$/HR: oxygen pulse, VD: dead space, VT: tidal volume, RER: respiratory exchange ratio, $PETCO_2$: end-tidal carbon dioxide pressure, $SpO_2$: oxygen saturation, SD: standard deviation.

*Significant p value < 0.05.

[$] Significance between this group and the others.

$VO_2$/HR (i.e., oxygen pulse) was significantly lower among ULB group versus both LL and control groups (p = 0.044, Table 2). However, there was no statistically significant difference regarding work load, $VO_2$, $VO_2$% predicted, $VCO_2$, peak $VO_2$, VE, breathing reserve, $PETO_2$, $PETCO_2$, VD/VT, respiratory rate, heart rate, RER and ventilation equivalent ($VE/VCO_2$ slope) at maximal exercise (p > 0.05, Table 2).

## ET programs

Tables 3 and 4 display the measures following ET compared to that before ET. There was no statistically significant change in FVC, FVC% predicted, $FEV_1$, $FEV_1$ / FVC before and after ET either in LL or ULB groups (p >0.05, Table 3). However, the FEV1% predicted significantly improved among ULB group after ET (55.2 ± 13.56% vs. 61.9 ± 12.07 before and after ET respectively, p = 0.035, Table 3). Further, $SpO_2$ significantly improved after ET training among ULB (median of 92.5% (IQR = 92.0–93.0) before vs. 93.5% (IQR = 91.0–94.0) after ET, p = 0.035, Table 3).

The absolute change in the average distance covered at the 6MWD after ET was significantly improved from 276.0 ± 130.06 meters to 438.0 ± 128.74 meters for LL group with a mean difference > 100 meters (p < 0.001, Table 3), and from 264.0 ± 95.13 meters to 414.0 ± 132.35 meters in ULB group with a mean difference > 100 meters (p < 0.001, Table 3). Notably, SGRQ score improved significantly in all its categories as well as total score after ET among both LL and ULB groups (p < 0.05, Table 3). Furthermore, the mean mMRC dyspnea

**Table 3. Comparison between spirometric parameters, SpO₂, 6MWT, and SGRQ before and after ET among group LL and ULB.**

| Test | LL group before ET (n = 10) | LL group after ET (n = 10) | Sig. (p)[#] | ULB group before ET (n = 10) | ULB group after ET (n = 10) | Sig. (p)[#] | Sig. (p)[$] |
|---|---|---|---|---|---|---|---|
| **Spirometry:** | | | | | | | |
| **FVC (L)** | 1.97 ± 0.60 | 2.01 ± 0.58 | 0.266 | 1.93 ± 0.53 | 2.16 ± 0.49 | 0.087 | 0.545 |
| **FVC (% predicted)** | 49.0 ± 12.30 | 53.9 ± 11.4 | 0.104 | 49.9 ± 11.89 | 56.5 (54.0–58.0) | 0.085 | 0.186 |
| **FEV₁ (L)** | 1.99 ± 0.58 | 2.34 ± 0.86 | 0.580 | 1.694 ± 0.38 | 2.10 ± 0.52 | 0.056 | 0.456 |
| **FEV₁ (% predicted)** | 55.7 ± 14.69 | 60.0 ± 16.18 | 0.247 | 55.2 ± 13.56 | 61.9 ± 12.07 | 0.035* | 0.778 |
| **FEV₁ / FVC** | 91.1 ± 9.15 | 95.6 ± 10.83 | 0.494 | 88.1 ± 8.43 | 88.1 ± 7.25 | 0.714 | 0.093 |
| **SpO₂ (%); median (IQR)** | 93.0 (92.0–96.0) | 94.0 (92.0–96.0) | 0.066 | 92.5 (92.0–93.0) | 93.5 (91.0–94.0 | 0.035* | 0.468 |
| **6MWD (meter)** | 276.0 ± 130.06 | 438.0 ± 128.74 | <0.001* | 264.0 ± 95.13 | 414.0 ± 132.35 | <0.001* | 0.686 |
| **SGRQ questionnaire (%, total); mean (±SD)** | 74.6 ± 16.73 | 26.7 ± 8.70 | <0.001* | 75.8 ± 18.78 | 32.5 ± 5.93 | <0.001* | 0.102 |
| **SGRQ (%, activity)** | 0.76 (0.69–0.76) | 37.4 (30.0–76.0) | 0.011* | 0.75 (0.74–0.76) | 44.9 (44.9–76.0) | 0.028* | 0.390 |
| **SGRQ (%, impact)** | 67.4 ± 17.33 | 10.7 ± 5.33 | <0.001* | 64.1 ± 17.26 | 11.3 ± 3.53 | <0.001* | 0.757 |
| **SGRQ (%, symptoms); median (IQR)** | 74.3 (70.7–78.1) | 54.4 (51.4–57.5) | 0.005* | 76.7 (72.4–77.2) | 57.5 (57.5–57.5) | 0.021* | 0.204 |
| **mMRC dyspnea scale** | 2.90 ± 0.74 | 2.0 ± 0.94 | 0.001* | 3.0 ± 0.67 | 1.8 ± 0.79 | <0.001* | 0.613 |

Abbreviations; FVC: forced vital capacity, FEV₁: forced expiratory volume in 1 second, 6MWD: 6-minute walk distance, SGRQ: St. George's Respiratory Questionnaire, SpO₂: oxygen saturation, mMRC dyspnea scale: modified medical research council dyspnea scale.

\* Significant *p* value < 0.05

$ Comparison between LL and ULB groups after ET

# Comparison between the same group after ET.

scale reduced significantly from 2.90 ± 0.74 to 2.0 ± 0.94 after ET (*p* = 0.001) in LL group, and from 3.0 ± 0.67 to 1.8 ± 0.79 (*p* <0.001) in ULB group.

VO₂, VO₂% predicted, VO₂/HR% predicted and SpO₂ at baseline (at 0 watts of WL) significantly improved after ET training among ULB group (*p* < 0.05, S2 Table in S1 File) but not after LL only training program (*p* > 0.05, S2 Table in S1 File). After training, at baseline CPET (0 watts of WL), resting HR in both groups were reduced and VT increased but not at a significant level (*p* > 0.05, S2 Table in S1 File). At maximal exercise, there was statistically significant increase of WL, WL% predicted, VE, VE% predicted, VT, and peak VO₂ (i.e., VO₂/kg at maximal WL) after ET in both groups (LL only and ULB) (*p* < 0.05, Table 4). Moreover, VO₂% predicted, VCO₂ and HR significantly increased among ULB training group (*p* < 0.05, Table 4) but not LL training group. However, neither ventilation equivalent (VE/VCO₂ slope) despite apparent decrease nor SpO₂ despite apparent increase had statistically significant difference after ET in both groups (*p* > 0.05, Table 4).

There were no statistically significant differences between LL training and ULB training programs regarding pulmonary function testing, SpO₂ at rest, 6MWT and SGRQ (*p* > 0.05, Table 3). Further, there was no statistically significant differences between LL and ULB training programs in terms of CPET parameters (*p* > 0.05, Table 4) except for VE and VE% predicted that was significantly higher in LL vs. ULB after termination of ET training sessions (*p* = 0.01 and 0.034 respectively, Table 4). We did not report any serious adverse events during any of the ET programs.

## Outcome

Regarding the control group, one patient (9.1%) died and 5 patients (45.5%) reported exacerbations of their underlying disease (Fig 1). Further, the patients of the control group did not

**Table 4. Comparison between CPET at maximal exercise before and after ET among group LL and ULB.**

| Variable | LL group before ET (n = 10) | LL group after ET (n = 10) | Sig. (p) [#] | ULB group before ET (n = 10) | ULB group after ET (n = 10) | Sig. (p) [#] | Sig. (p) [$] |
|---|---|---|---|---|---|---|---|
| Work load (watts) | 49.20 ± 14.48 | 64.7 ± 11.51 | 0.009* | 37.88 ± 21.62 | 51.2 ± 20.36 | 0.006* | 0.085 |
| Work load% | 33.7 ± 14.94 | 45.8 ± 14.66 | 0.038* | 30.4 ± 19.32 | 38.3 ± 19.98 | 0.048* | 0.353 |
| VE (l/min) | 44.18 ± 11.50 | 54.67 ± 8.72 | 0.017* | 31.81 ± 7.78 | 44.47 ± 7.0 | 0.003* | 0.010* |
| VE% | 42.1 ± 14.87 | 53.6 ± 10.16 | 0.021* | 33.0 ± 9.38 | 44.9 ± 5.96 | 0.005* | 0.034* |
| Breathing reserve (%) | 46.5 ± 16.59 | 53.0 ± 10.79 | 0.284 | 58.30 ± 16.99 | 58.7 ± 4.85 | 0.936 | 0.145 |
| VT | 0.98 ± 0.21 | 1.16 ± 0.24 | 0.037* | 0.82 ± 0.19 | 1.08 ± 0.31 | 0.017* | 0.538 |
| VD/VT ratio | 0.22 ± 0.06 | 0.22 ± 0.03 | 0.726 | 0.21 ± 0.05 | 0.23 ± 0.04 | 0.112 | 0.296 |
| $VO_2$ (L/min) | 1.19 (0.94–1.39) | 1.34 (1.04–1.86) | 0.169 | 0.79 (0.56–1.10) | 0.94 (0.88–1.06) | 0.139 | 0.140 |
| $VO_2$% | 63.0 (46.0–72.0) | 76 (57.0–85.0) | 0.066 | 36.5 (29.0–46.0) | 49.5 (39.0–60.0) | 0.022* | 0.050 |
| $VO_2$/kg (ml/kg/min) | 17.5 (13.0–23.0) | 22 (17.0–24.0) | 0.032* | 10.5 (5.0–16.0) | 13.5 (11.0–21.0) | 0.018* | 0.075 |
| $VCO_2$ (L/min) | 0.93 (0.64–1.03) | 1.09 (0.95–1.26) | 0.051 | 0.63 (0.36–0.68) | 0.87 (0.74–1.09) | 0.009* | 0.120 |
| Respiratory rate (br/min) | 47.50 ± 7.89 | 46.55 ± 7.86 | 0.695 | 40.0 ± 9.38 | 43.63 ± 7.58 | 0.240 | 0.409 |
| RER | 0.81 (0.72–0.83) | 0.74 (0.67–0.84) | 0.507 | 0.80 (0.64–0.89) | 0.83 (0.67–0.88) | 0.540 | 0.402 |
| HR (b/min) | 140 (122.25–144) | 140.5 (119.5–147.0) | 0.715 | 122 (103.5–139.5) | 130 (127.0–145.5) | 0.046* | 0.705 |
| HR% | 78 (69.2–84.8) | 80.8 (66.9–85.4) | 0.715 | 69 (65.0–82.3) | 80.1 (74.0–84.5) | 0.063 | 1.00 |
| SBP (mmHg) | 138.8 ± 7.74 | 138.5 ± 9.73 | 0.879 | 143 ± 10.01 | 143 ± 7.89 | 1.0 | 0.271 |
| DBP (mmHg) | 93.5 ± 10.01 | 94.5 ± 8.64 | 0.509 | 98.5 ± 10.56 | 98.5 ± 7.09 | 1.0 | 0.273 |
| $VE/VCO_2$ | 40.85 ± 14.22 | 38.35 ± 14.69 | 0.265 | 45.83 ± 10.96 | 38.14 ± 9.84 | 0.097 | 0.477 |
| $VO_2$/HR (ml/beat) | 8.7 (6.23–10.15) | 7.5 (6.9–8.3) | 0.715 | 3.9 (3.80–6.08) | 6.9 (5.78–7.98) | 0.128 | 0.390 |
| $VO_2$/HR% | 70.4 (0.56–0.85) | 73.5 (66.4–82.3) | 1.00 | 47.9 (0.45–0.64) | 74.7 (59.0–86.9) | 0.128 | 0.705 |
| $PETCO_2$ (mmHg) | 24.5 (23.0–30.0) | 25 (22.0–29.0) | 0.610 | 27.0 (23.0–33.0) | 27.5 (25.0–29.0) | 0.812 | 0.518 |
| $PETO_2$ (mmHg) | 116.0 (106.0–121.0) | 120.5 (113.0–127.0) | 0.858 | 122.0 (113.0–126.0) | 116.5 (111.0–121.0) | 0.138 | 0.306 |
| $SpO_2$ (%) | 86.8 ± 5.25 | 89.3 ± 5.50 | 0.179 | 87.8 ± 5.2 | 87.5 ± 5.76 | 0.883 | 0.484 |

Abbreviations; VE: minute ventilation, br/min: breath/minute, $VO_2$: oxygen consumption, $VCO_2$: carbon dioxide output, HR: heart rate, $VO_2$/HR: oxygen pulse, VD: dead space, HR: heart rate, VT: tidal volume, RER: respiratory exchange ratio, $PETCO_2$: end-tidal carbon dioxide pressure, $SpO_2$: oxygen saturation, SBP: systolic blood pressure, DBP: diastolic blood pressure, SD: standard deviation

* significant p value < 0.05

$ Comparison between LL and ULB groups after ET

# Comparison between the same group after ET.

report change of their dyspnea level or other associated symptomatology during their follow up through phone calls despite of continuing medical therapy prescribed by their physicians for at least 3 months. Regarding the ET groups, none of the patients died while one patient (5%) experienced exacerbation of symptoms during the follow-up duration (Fig 1). Moreover, ET (either LL only or ULB) was an independent significant protective factor against underlying disease exacerbations or mortality (p = 0.020, OR = 0.063, CI95% = 0.006–0.652) after adjusting to age, gender, smoking status, and corticosteroids use (S3 Table in S1 File).

## Discussion

We found that the f-ILD patients, regardless of the etiology, whom were subjected to ET either LL or ULB training program had improved in term of functional capacity as being assessed by CPET and 6MWT as well as HRQoL. There was no significant improvement of FVC; however, dyspnea level and $SpO_2$ significantly improved after ET. Further, we did not find significant

difference between LL and ULB training programs regarding the follow-up assessment except for peak VE.

Holland et al [27] found that 6MWD and dyspnea improved significantly after ET without significant difference between the IPF patients and non-IPF. Vainshelboim et al [23] found in their clinical trial that 6MWD, dyspnea, quality of life, peak $VO_2$ and work rate assessed by CPET improved significantly in IPF population after ET. Also, Kozu et al [28] and Dowman et al [29] found that dyspnea, 6 MWD and quality of life improved in IPF and other ILD after ET. Similarly, Perez-Bogerd et al [30] found in their cohort of ILD that 6MWD, SGRQ and peak work rate increased significantly after PR. Our results are in accordance with these findings.

Treatment options for ILD are limited. Available drug therapy has significant toxic side effects and may not be suitable for many patients with no evidence that current drug therapies for f-ILD can improve quality of life and symptoms [31]. Interestingly, the 6MWD in the current study exceeded the increases observed in most of the previous studies (>100 m vs. 25–45 m respectively) [23, 27–30]. This could be explained by patients' motivation and the adherence to the ET sessions as well as the lesser proportion of participants diagnosed with IPF who experienced minimal change in 6MWD in other studies [23, 27–30].· Further, the younger age in the ET groups rather than the control group (Table 1) could be another factor that encouraged the participants to stuck to our ET programs.

In contrast to the study of Vainshelboim et al [23], we did not report improvement in pulmonary functions especially FVC after both modalities of ET. However, Holland et al [27] did not find improvement of FVC of their studied IPF population subjected to PR program, similar to our results. This difference could be attributed to the heterogeneity of our f-ILD population, the severity of the disease and the difference of the duration of the ET provided as well as the various training programs in the studies. Up to date, there is no standardized ET recommended for ILD patient and various programs of ET were applied in clinical trials [11, 23, 27, 29, 32]. Further, pulmonary function did not typically improve after PR in other respiratory diseases as COPD which could not be considered as primary outcome in ET programs.

Further, beside the significant improvements in peak $VO_2$, the gold standard for cardiorespiratory capacity evaluation, we showed also significant improvements in WL, VE, and VT at maximal exercise as well as $SpO_2$ at rest. In contrast to our findings, Holland et al [27] and Arizono et al [33] did not report difference in peak $VO_2$ after ET, but they found significant improvements of other CPET parameters in their IPF patients. The effect of ET on improvement of physiological outcomes as detected with CPET and the clinical outcome as being reflected by HRQoL and dyspnea improvements in our patients can be explained by several mechanisms. Firstly, repetitive stimulation of high ventilatory demands and stretching of the thoracic muscles during ET sessions as well as chest expansion during exercises resulted in efficient breathing, improvement of respiratory muscles strength, enhancement of the pleural elasticity and pulmonary compliance resulting in increase of peak VE and VT [23, 34–37] and so amelioration of dyspnea perception. Secondly, enhancement of the ventilatory responses that occurred after the ET could be a cause of recruitment of more alveoli and so increased alveolar oxygen tension and improved alveolar ventilation / perfusion mismatch, resulting in increasing peak $VO_2$ [34, 35].

We have shown that ET with the targeted intensity reached during the training sessions indicate that ET is safe and feasible to be implemented in f-ILD in a similar way as in COPD and other chronic respiratory condition [38]. Moreover, in the current study we demonstrated 2 modalities of ET (LL only and ULB). To our knowledge, this is the first study that evaluates 2 different training programs in f-ILD patients. Interestingly, we found no clinically significant important difference between the two groups. This had the importance of implication of ET

with only LL program which would be time and effort saving, especially for f-ILD patients whom have a low exercise tolerance, and in our developing country that has limited resources.

## Limitations

The current study had some limitations. Firstly, we did not assess the peripheral muscle strength in the current study. Peripheral muscle weakness is predictive of exercise limitation and intolerance in ILD and further studies still required to assess this factor [39]. Secondly, the current study included only one component of PR, the ET, as we did incorporate the educational and nutritional components which has been shown to be associated with comparable even greater clinical outcomes compared with ET alone in some studies [26]. However, still ET constitutes the main bulk of all PR programs, as reported in previous studies of COPD patient [23, 40]. Thirdly, we did not provide objective follow-up of the control group (using either CPET or 6MWT) and we considered only subjective follow-up through phone calls. However, dyspnea assessed by mMRC was important predictor of mortality among chronic ILD either the baseline level or longitudinal increases of dyspnea degree [41]. Fourthly, we did not include the diffusion capacity (DLCO) in the evaluation of our participant due to lack of this facility in our institute. DLCO is crucial for ILD evaluation; however, FVC variability is associated with disease progression and widely accepted as single factor for monitoring of disease in clinical trials [42, 43]. Lastly, the sample size of the current study is quietly small which could limit the external validity, increase the risk of statistical error and did not allow to conduct power analysis despite the high debate regarding power analysis validity in case of significant data [44, 45]; so further studies are still needed to confirm the current results.

## Conclusions

ET in f-ILD is safe, tolerable and results in improving the functional exercise capacity, dyspnea, oxygen saturation, and HRQoL which highlights the effective of ET for f-ILD population. Further, LL only training program is effective as ULB program. This can be of advantage especially in low economic countries that had low resources and to decrease the effort needed by those patients with similar results.

## Supporting information

**S1 Checklist.**
(DOC)

**S1 File. Supplemental material.** The file contains methods, S1-S3 Tables and S1, S2 Figs legends.
(DOCX)

**S1 Protocol.**
(PDF)

## Author Contributions

**Conceptualization:** Nashwa Hassan Abdel Wahab, Gihan Younis, Enas El-sayed, Hanaa Shafiek.

**Data curation:** Hatem Essam.

**Formal analysis:** Hatem Essam, Hanaa Shafiek.

**Investigation:** Hatem Essam, Nashwa Hassan Abdel Wahab, Gihan Younis, Hanaa Shafiek.

**Methodology:** Hatem Essam, Nashwa Hassan Abdel Wahab, Gihan Younis, Hanaa Shafiek.

**Software:** Hatem Essam.

**Supervision:** Nashwa Hassan Abdel Wahab, Gihan Younis, Enas El-sayed, Hanaa Shafiek.

**Validation:** Nashwa Hassan Abdel Wahab, Gihan Younis, Enas El-sayed, Hanaa Shafiek.

**Visualization:** Nashwa Hassan Abdel Wahab, Gihan Younis, Enas El-sayed, Hanaa Shafiek.

**Writing – original draft:** Hatem Essam, Hanaa Shafiek.

**Writing – review & editing:** Nashwa Hassan Abdel Wahab, Gihan Younis, Enas El-sayed, Hanaa Shafiek.

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
