## [Decision Letter · Decision Letter 0]

10 Mar 2022

PONE-D-21-37890Effects of different exercise training programs on the functional performance in fibrosing interstitial lung diseases: a randomized trialPLOS ONE

Dear Dr. Shafiek,

Thank you for submitting your manuscript to PLOS ONE. After careful consideration, we feel that it has merit but does not fully meet PLOS ONE’s publication criteria as it currently stands. Therefore, we invite you to submit a revised version of the manuscript that addresses the points raised during the review process.

We look forward to receiving your revised manuscript.

Kind regards,

Walid Kamal Abdelbasset, Ph.D.

Academic Editor

PLOS ONE

Journal Requirements:

Reviewers' comments:

Reviewer's Responses to Questions

**Comments to the Author**

1. Is the manuscript technically sound, and do the data support the conclusions?

Reviewer #1: Yes

Reviewer #2: Partly

Reviewer #3: Partly

2. Has the statistical analysis been performed appropriately and rigorously? 

Reviewer #1: Yes

Reviewer #2: N/A

Reviewer #3: Yes

3. Have the authors made all data underlying the findings in their manuscript fully available?

Reviewer #1: Yes

Reviewer #2: Yes

Reviewer #3: Yes

4. Is the manuscript presented in an intelligible fashion and written in standard English?

Reviewer #1: Yes

Reviewer #2: Yes

Reviewer #3: Yes

5. Review Comments to the Author

Reviewer #1: The authors conduct a case-control study to compare the effects of different exercise training programs (two exercise programs: LL endurance training, and ULB program, and one control group) on the functional performance in fibrosing interstitial lung diseases with 31 subjects. The results show that exercise training programs were associated with improved exercise capacity and quality of life.

1. Line 96. 11 patients who refused to participate in ET programs or unable to participate due to certain reasons were considered as a control group. In other words, the selection of control groups were not randomized. How do we know the significant results from intervention or other reasons? Such self-selection and non-randomization seems to violate the spirit of randomized trial.

2. Line 163. There were significant difference in ET group and control at baseline. Were these significant covariates adjusted in later analysis? If not, the final results or conclusions might be questionable in terms of generalizability?

Reviewer #2: Thank you for your submission. This kind of long term studies are challenging and necessary to advance the field. I appreciate the hard work and effort and a well written article in all aspects. But I think there is problem in methodology especially in standardisation of groups and risk of selection bias. I'm undecided in accepting this article, I'm half and half.

I believe these major revisions below will make the article much more effective for the readers:

Comments

1) In abstract section,

Only 18 sessions were written as exercises. İf it is possible, the number of days, whether there is a break, etc. should be emphasized more clearly in the abstract and material method section.

2) The introduction section,

is well written. The purpose is well emphasized.

3) Method section:

Write the population age groups more clearly, is it the elderly group or the adults?

How did you measure exercise intensity with? Please explain it.

Increasing exercise intensity and durations should have been more standardized, it seems not objective and clear. Please explain week by week.

4) Statistical analysis section,

Power analysis should be done, the strength of the study should be specified, such as ( post hoc analysis )

5) Results section,

include bias risk (corticosteroid use). It is risky to include users and non-users in the same group or comparative groups in the results.

6) Please check the references like (28) Is it necessarry to highlight it?

7) Table section,

Please delete the dollar sign under the control group in Table 1.

The age difference between the control group and the experimental groups is very large.And also it is obvious thatcontrol group are in worse physical condition than other groups.

Reviewer #3: � Abstract:

In my opinion, unit forVO2should be added.

In methods section :

I thought it would be better if images about aerobic exercise tests were used ( the monitor of the device, etc.)

DLCO is crucial in interstitial lung diseases. Is there any particular reason for not mentioning about it?

The literature or information about mMRC dispnea scale should be added.

InStatistical analysis:

Line 153‘All the data were expressed as median and interquartile range (IQR) or mean . standard deviation(SD) according to the normal distribution of continuous data’.

This sentence hasa missing part. “Median and interquartile rangefor non-normal distribution“should be added.

In the Results:

Table 1 has both weight and BMI values. In my opinion, emphasizing “weight” as a separate entity is not necessary. Also, “Y/N” statements for comorbidities have only Y statement in the table. So, “N” statement should be removed.

In Table 1CPEE is considered to be a phenotype of IPF, so it can be merged.

182

“The VO2, VO2% predicted,VO2/kg and oxygen pulse (VO2/HR) at baseline were significantly lower among ULB groups compared to LL group and control group (p < 0.05, table S1 online supplement)”Could you please check this sentence? Only VO2/HR is statistically significant according to Table 1.

184-185“At maximal exercise, SpO2 of the control group was significantly lower when…”Isn’t the control group selected from the ones do not exercise??

186-187 ‘while the oxygen pulse was significantly lower among ULB group

versus both LL and control groups (p= 0.045, table 2)’ I didn’t see these results in table 2.

210-211‘VO2, VO2% predicted, VO2/HR% predicted and SpO2 at baseline significantly improved after ET training among ULB group (p < 0.05, table S2, online supplement) but not after LL only training program (p > 0.05, table S2, online supplement)’.

In table 4, VO2 and VO2/HR%, are not statistically significant for ULB group.

212-213

‘After training, resting HR in both groups were reduced and VT increased but not at a significant level (p > 0.05, table S2, online supplement’

Could you please check this sentence? VT is decreased significantly in table 4. HR results for both groups are different.

215- ‘peak VO2 after ET in both groups (LL only and ULB) (p < 0.05, table 4)’ Could you pleasecheck this sentence? This statement is not statistically significant according to table 4.

6. PLOS authors have the option to publish the peer review history of their article (what does this mean?). If published, this will include your full peer review and any attached files.

Reviewer #1: No

Reviewer #2: **Yes: **Esedullah AKARAS

Reviewer #3: No

---

## [Author Response · Author response to Decision Letter 0]

24 Apr 2022

Dear Dr. Walid Kamal Abdelbasset; Academic Editor

I would like to thank the editor in-chief, the academic editor, the editorial office, and the reviewers for your time and effort in reviewing the paper and the valuable comments. We truly think that the manuscript has improved sensibly after your suggestions. We would like to respond all reviewer’s comments point by point and fulfil all the journal requirements.

Journal Requirements:

Response. The style of manuscript has been ajusted to meet PLOS ONE's style requirements. 

2. We note that you have indicated that data from this study are available upon request. PLOS only allows data to be available upon request if there are legal or ethical restrictions on sharing data publicly. 

Response. After reviewing the legal or ethical restrictions on sharing data publicly, we included the dataset of the current study in one the public repository (Fig share). URL https://figshare.com/s/fd724ded44c70a86cef5 and doi: 10.6084/m9.figshare.19640214.

The dataset is currently as a private link that will be available as public on acceptance. We provided this point in a new cover letter submitted with the revised version. 

Reviewer #1: 

Thank you for your valuable evaluation and your time.

1. Line 96. 11 patients who refused to participate in ET programs or unable to participate due to certain reasons were considered as a control group. In other words, the selection of control groups were not randomized. How do we know the significant results from intervention or other reasons? Such self-selection and non-randomization seems to violate the spirit of randomized trial. 

Response. The benefits of rehabilitation and exercise training (ET) in patients with ILD in terms of improvement of functional exercise capacity and quality of life was previously shown in various studies (Dowman LM et al. The evidence of benefits of exercise training in interstitial lung disease: a randomised controlled trial. Thorax. 2017;72(7):610-9; Huppmann P et al. Effects of inpatient pulmonary rehabilitation in patients with interstitial lung disease. Eur Respir J. 2013;42(2):444-53). So, we think that it is unethical to choose a randomized control group among our studied population and we have a specialized ET program of care for ILD patients. Further, the aim of the current study was to compare the effects of different ET programs on the respiratory performance, exercise capacity and quality of life in progressively fibrosed ILD patients rather than to explore the general effect of ET in ILD patients compared to usual care. 

Accordingly, we offered our ET program to all participants and those who refused to participate in ET programs or unable to participate due to morbid obesity or living outside the influence of the hospital or voluntary withdrew from the study as we mentioned was considered as control; while those who agreed to participate in ET programs were totally randomized to one of ET (LL only or ULB program) who were included in the subsequent analysis later and we assure that there was no self-selection or non-randomization among the interventions groups. Moreover, the inclusion and exclusion criteria for 3 groups were the same with no statistical difference regarding duration of disease, associated comorbidities, underlying pathological ILD disease and severity of lung fibrosis (assessed by spirometry) as we showed in table 1; so, we think that the significant results (provided in tables 3-4 and S1 file online supplement) were related to the provided intervention rather than the influence of other factors. We clarified this point in the methodology. (Line 104 -109)

2. Line 163. There were significant difference in ET group and control at baseline. Were these significant covariates adjusted in later analysis? If not, the final results or conclusions might be questionable in terms of generalizability? 

Response Following the reviewer’s recommendation, we conducted a multivariate logistic regression adjusted to the significant baseline covariates shown in table 1 and corresponding text (namely age, gender, smoking status, corticosteroids) using forward method. We found that only ET (either LL only or ULB) was an independent significant protective factor against underlying ILD disease exacerbations or mortality (p= 0.020, odd ratio= 0.063, confidence interval 95%= 0.006 – 0.652) after adjusting to age, gender, smoking status, and corticosteroids use which were insignificant in regression model. Accordingly, this data supports our conclusion regarding the benefits of ET among f-ILD and reject the terms of generalizability. We included this part in the results section (outcome section, Lines 291 - 294) and online supplement as table S3 with explanation of statistical method in methodology (Lines 205 - 208). 

Reviewer #2: 

Thank you for your valuable evaluation and your time.

1) In abstract section,

Only 18 sessions were written as exercises. İf it is possible, the number of days, whether there is a break, etc. should be emphasized more clearly in the abstract and material method section. 

Response. Following the reviewer’s recommendation, we emphasized more clearly the exercise sessions in the abstract in terms of number of days and weeks as we had 3 supervised sessions of exercise training / week for total duration of 6 consecutive weeks (total number 18 sessions). And we provided details of the exercise training protocol in the methodology as being suggested later by the reviewer. (Line 29 in Abstract and Lines 149 – 150 in methods) 

2) The introduction section,

is well written. The purpose is well emphasized.

Response. We would like to thank the reviewer for his comment.

3) Method section:

Write the population age groups more clearly, is it the elderly group or the adults?

How did you measure exercise intensity with? Please explain it.

Increasing exercise intensity and durations should have been more standardized, it seems not objective and clear. Please explain week by week. 

Response. Regarding the population age groups, we included adult patients not elderly groups. The mean age of our studied population was 40s (as we showed in table 1), with age range between 25 to 70 years, rather than older than 65 years which represented the elderly group. We clarified this point in the methodology and we provided the age range of the studied population. (Line 99)

Regarding the exercise intensity, it was measured as a percentage of maximum heart rate determined from the equation (220 – age of the participant). We clarified this point in the methodology (Lines 154 – 155) and supported by reference 24.

Following the reviewer’s recommendation, we provided a detailed description of ET program in the methodology week by week. Our ET program followed the standards of exercise prescription were applied as previously described for chronic lung diseases (Bolton CE, et al. British Thoracic Society guideline on pulmonary rehabilitation in adults. Thorax. 2013;68 Suppl 2:ii1-30). The program was individualized; as the initial duration, the initial intensity, and the rate of progression varied among patients based on their exercise tolerance which is widely accepted by previous publication (Dowman LM, et al. The evidence of benefits of exercise training in interstitial lung disease: a randomised controlled trial. Thorax. 2017 Jul;72(7):610-619; Vainshelboim B, et al. Exercise training-based pulmonary rehabilitation program is clinically beneficial for idiopathic pulmonary fibrosis. Respiration. 2014;88(5):378-88.) During 1st week, the patients exercised initially at low intensity exercise i.e., 50-60% of their maximum heart rate and short duration of usually 10 minutes that was broken into shorter intervals if needed (as cycles of 3 minutes training followed by 1-2 minutes of rest period). The 2nd week, an attempt was made to increase the performed work during training by increasing the duration of session by 5 minutes every 2 sessions with decreasing the intervals between training, and increasing the workload by 5% every 2 – 3 sessions according to patient’s tolerance. The 3rd week, most patients were able to continue 30 minutes of aerobic exercise as 2 cycles of continuous 15 minutes aerobic training separated by one interval of rest at moderate exercise intensity of 64 –76% of their maximum heart rate. The 4th – 6th week, the ET continued the achievement of 3rd week whereas most patients were able to exercise 30 minutes continuously at moderate exercise intensity which was the main target to achieve. This part was added with details in the methodology section (ET program subtitle, Lines 151 – 169 and references 22 – 24) and supported with figures in the supplemental material (S1 file, fig S1 and S2) as being suggested by the reviewer #3.

4) Statistical analysis section,

Power analysis should be done, the strength of the study should be specified, such as (post hoc analysis)

Response. We agree with the reviewer that power analysis as post hoc analysis increase the strength of the study; however, post-hoc analysis is usually conducted on negative data (insignificant p value) as a method to show that a “non-significant” hypothesis test failed to achieve significance because it wasn’t powerful enough. Further, it is well documented that post hoc power calculations are not useful and could be misleading (Goodman and Berlin. The use of predicted confidence intervals when planning experiments and the misuse of power when interpreting results. Ann Intern Med. 1994 Aug 1;121(3):200-6; Hoenig and Heisey. The Abuse of Power: The Pervasive Fallacy of Power Calculations for Data Analysis. The American Statistician, February 2001, Vol. 55, No. 1: 19-24; Althouse AD. Post Hoc Power: Not Empowering, Just Misleading. J Surg Res, 2021; 259: A3-A6). In the current study, there was statistically significant improvement in health quality-of-life evaluation, reduction of dyspnea (as assessed by mMRC), improvement of respiratory performance and exercise capacity that was assessed by CPET and 6-minute walk test (p <0.05, tables 3-4 and S2) and only few parameters were not significant during the analysis which we think did not influence the power of the study. However, following the reviewer’s recommendations, we calculated post-hoc power on one of insignificant parameters; and we found that achieved statistical power 60% to detect 3% in the mean improvement of maximal SpO2 after ET between 2 groups at level of significance of 0.05 using 2-sided independent t test (data not shown). We assumed that the power of 60% was due to low sample size of our study; however, our sample size is based on the prevalence of the ILD disease in our community (Shafiek H, et al. Transbronchial cryobiopsy validity in diagnosing diffuse parenchymal lung diseases in Egyptian population. J Multidiscip Healthc. 2019 Aug 30;12:719-726; El-Hoffy MM, et al. High resolution multi-detector row computed tomography in imaging of interstitial lung diseases. Alexandria J Med. 2008;44(2):1–7) as well as the relatively uncommon f-ILD worldwide and its high variability (Kaul B, et al. Variability in Global Prevalence of Interstitial Lung Disease. Front Med (Lausanne). 2021;8:751181). We highlighted this point and relatively small sample size in the limitation of the study. (Lines 369-373)

5) Results section,

include bias risk (corticosteroid use). It is risky to include users and non-users in the same group or comparative groups in the results.

Response. We agree with the reviewer that inclusion of users and non-users of corticosteroids in the groups could raise a bias risk; however, as we showed in table 1 that 90% of the participants in ET programs were on oral corticosteroids on inclusion in the study which was equal between both exercise training groups as the significant difference was between the control group and the experimental groups (as we shown in table 1), with no statistically significant difference regarding the duration of the disease in the studied groups. Further, as being suggested by reviewer #1, we conducted a multivariate regression analysis with adjustment to all significant baseline covariates and we found that corticosteroids use did not influence the benefits of ET regarding the outcome. In addition, our study focused on patients with fibrotic progressive ILD, where the rule of corticosteroid use is variable and questionable (Richeldi L et al. European Respiratory Review Dec 2018, 27 (150) 180074; DOI: 10.1183/16000617.0074-2018); and we did not modify any line of the treatment was considered by the patients at the time of inclusion to our programs in order to avoid any bias in our results. Accordingly, we think that there was no bias risk in the current results. We highlighted this point in the methodology (Lines 113 – 116, and 205 – 208) and results (lines 291 – 294, table 1, table S3 – supplemental material file S1).

6) Please check the references like (28) Is it necessarry to highlight it?

Response. We agree with the reviewer that it is not necessary to highlight reference 28. The reference has been corrected and became reference 30.

7) Table section,

Please delete the dollar sign under the control group in Table 1.

The age difference between the control group and the experimental groups is very large. And also it is obvious thatcontrol group are in worse physical condition than other groups. 

Response. Following the reviewer’s recommendation, we deleted the dollar sign under the control group in Table 1 and its comment in the table note. 

Secondly, we agree with the reviewer that the age group is large and there is significant age difference between the control group and the experimental groups; however, the control group is not in worse physical condition than the experimental groups based on the insignificant difference between various groups regarding the pulmonary function test (spirometry), health related quality of life evaluation, 6-minute walk distance achieved and CPET data that we described in tables 1 and 2. Further and as being suggested by reviewer #1, we conducted a multivariate analysis model with adjustment to all significant baseline covariates and we found that age did not influence the benefits of ET regarding the outcome. In addition, we think that the younger age of the experimental groups could be an important factor that encouraged the participants to stuck to our exercise training programs. We highlighted this point in the methods (Lines 205 – 208), results (Lines 291 – 294, table 1, table S3 – supplemental material file S1) and discussion (Lines 317 – 319). 

Reviewer #3: 

Thank you for your valuable evaluation and your time.

Abstract: 

- In my opinion, unit for VO2 should be added. 

Response. Following the reviewer’s recommendation, we included the unit change for VO2 in both exercise training modalities in abstract (Lines 36 – 40). 

In methods section: 

- I thought it would be better if images about aerobic exercise tests were used (the monitor of the device, etc.) DLCO is crucial in interstitial lung diseases. Is there any particular reason for not mentioning about it? 

Response. Following the reviewer’s recommendation, we included images of the aerobic exercise tests (lower and upper limbs training) provided to our participants along with the monitoring device used mainly in case of LL training in the File S1 supplemental material (Figures S1 – S2). Secondly, we totally agree with the reviewer that DLCO is crucial in interstitial lung diseases; however, due to lack of this test in our institute during the duration of the study, we did not mention this test in our study. However, FVC variability is associated with disease progression (Veit T, et al. Variability of forced vital capacity in progressive interstitial lung disease: a prospective observational study. Respir Res. 2020 Oct 19;21(1):270. doi: 10.1186/s12931-020-01524-8) and FVC decline was considered as single primary end-point for response to therapy among progressive fibrotic ILD patients in clinical trials (Behr J et al. Pirfenidone in patients with progressive fibrotic interstitial lung diseases other than idiopathic pulmonary fibrosis (RELIEF): a double-blind, randomised, placebo-controlled, phase 2b trial. Lancet Respir Med. 2021 May;9(5):476-486. doi: 10.1016/S2213-2600(20)30554-3.). We highlighted this point in the limitation of the study (Lines 366 – 369, references 43 – 44). 

- The literature or information about mMRC dispnea scale should be added. 

Response. Following the reviewer’s recommendation, we added a reference about mMRC dyspnea scale (reference 17) with detailed information about mMRC dispnea scale in the supplemental material (S1 file) with a highlighted note in the main text (Line 111).

In Statistical analysis: 

- Line 153‘All the data were expressed as median and interquartile range (IQR) or mean. standard deviation(SD) according to the normal distribution of continuous data’.

This sentence hasa missing part. “Median and interquartile rangefor non-normal distribution“should be added. 

Response. We agree with the reviewer that this part “median and interquartile range for non-normal distribution” was missing and has been added. (Lines 200 – 201)

In the Results: 

- Table 1 has both weight and BMI values. In my opinion, emphasizing “weight” as a separate entity is not necessary. Also, “Y/N” statements for comorbidities have only Y statement in the table. So, “N” statement should be removed.

Response. We agree with the reviewer that emphasizing “weight” as a separate entity is not necessary, so we removed the weight from table 1. Also, we agree that “N” statement in “Y/N” statements for comorbidities is not necessary and we removed “N” for comorbidities in table 1 as being recommended by the reviewer.

- In Table 1 CPEE is considered to be a phenotype of IPF, so it can be merged.

Response. We agree with the reviewer that CPEE is considered to be a phenotype of IPF, so we merged it with IPF (in table 1) as being recommended by the reviewer and we re-calculated the p value. (Table 1)

- 182 “The VO2, VO2% predicted, VO2/kg and oxygen pulse (VO2/HR) at baseline were significantly lower among ULB groups compared to LL group and control group (p < 0.05, table S1 online supplement)” Could you please check this sentence? Only VO2/HR is statistically significant according to Table 1. 

Response. In table S1 of the supplemental material, we provided the baseline CPET parameters that was recorded before starting the incremental increase in work load of CPET (i.e., during the warming stage of CPET at workload of zero watts). Accordingly, we found that the VO2, VO2% predicted, VO2/kg and oxygen pulse (VO2/HR) at baseline (at 0 watts of workload) were significantly lower among ULB groups compared to LL group and control group (as we showed in the main text). However, only VO2/HR is statistically significant at maximal workload of CPET among ULB groups compared to LL group and control group as we shown in table 2 (not table 1). We clarified this point in the methodology with the addition of a description of baseline phase of CPET in order to avoid conflicts regarding the data provided (Lines 136 – 138) as well as short description in the heading of table S1 (supplemental material S1 file) and in results section (Lines 234 – 236). 

- 184-185 “At maximal exercise, SpO2 of the control group was significantly lower when...”Isn’t the control group selected from the ones do not exercise??

Response. Yes, the control group were selected from the ones who did not exercise training as mentioned by the reviewer; however, all the groups including the control group had the CPET on recruitment in the study as we meant by “At maximal exercise” the maximal workload during CPET. We corrected the word “maximal exercise” to “maximal work load during CPET” to avoid conflicts. (Line 238)

- 186-187 ‘while the oxygen pulse was significantly lower among ULB group

versus both LL and control groups (p= 0.045, table 2)’ I didn’t see these results in table 2. 

Response. In table 2, the oxygen pulse (which is VO2/HR) was significantly lower among ULB group (median of 3.9 (IQR= 3.80 – 5.65) ml/beat) versus both LL (median of 8.7 (IQR= 6.85 – 10.0) ml/ beat) and control groups (median of 10.2 (IQR= 4.1 – 10.4) ml/beat) and p value of 0.044 rather than 0.045 as written in the text. This point has been clarified in the text and p value was corrected. (Lines 239 – 240) 

- 210-211 ‘VO2, VO2% predicted, VO2/HR% predicted and SpO2 at baseline significantly improved after ET training among ULB group (p < 0.05, table S2, online supplement) but not after LL only training program (p > 0.05, table S2, online supplement)’.

In table 4, VO2 and VO2/HR%, are not statistically significant for ULB group.

Response. In table S2 of online supplement, we provided a comparison between ET groups regarding the baseline CPET parameters that was recorded during the warming phase of CPET with a workload of zero watts, and before starting the incremental increase of workload. Accordingly, the baseline variables (VO2, VO2% predicted, VO2/HR% predicted and SpO2 measured at 0 watts of workload) significantly improved after ET training among ULB group but not after after LL only training program. However, in table 4, we provided the CPET at maximal workload achieved by the participants during CPET and as was mentioned by the reviewer neither VO2 nor VO2/HR% were statistically significant for ULB group. We clarified this point in the methodology with the addition of a description of baseline phase of CPET (Lines 136 – 138) in order to avoid conflicts regarding the data provided as well as short description in the heading of table S2 (supplemental material S1 file) and in results section (Lines 267 – 269). 

- 212-213 ‘After training, resting HR in both groups were reduced and VT increased but not at a significant level (p > 0.05, table S2, online supplement’

Could you please check this sentence? VT is decreased significantly in table 4. HR results for both groups are different. 

Response. As being suggested by the reviewer, we checked the sentence in lines 212-213 and we found the data is correct with no contradictory between the data in table S2 and table 4. The resting HR in both groups (ULB and LL) were reduced and VT increased but not at a significant level at baseline CPET (i.e., workload of zero watts) as being shown in table S2 (supplemental material S1 file); however, VT is increased significantly after ET training among both groups as shown in table 4 and prescribed in the text. Regarding HR, the results for both groups are different as being mentioned by the reviewer whereas HR at maximal exercise among ULB increased significantly (p= 0.046, table 4) but not after LL training (p= 0.715, table 4). We clarified this point in the text (Lines 269 – 270) and we added a description of baseline phase of CPET in methodology (Lines 136 – 138). 

- 215- ‘peak VO2 after ET in both groups (LL only and ULB) (p < 0.05, table 4)’ Could you please check this sentence? This statement is not statistically significant according to table 4. 

Response. As being suggested by the reviewer, we checked the sentence in line 215. We agree with the reviewer that the word “peak VO2” could have some conflict, as we meant by the word “peak VO2” as VO2/kg at maximal work load which is statistically significant after ET in LL and ULB groups according to table 4 (p= 0.032 and 0.018 respectively). We clarified this point in the text (result section, Line 272) and we provided also a clear definition of peak VO2 in the methodology (Lines 139 – 140).

---

## [Decision Letter · Decision Letter 1]

3 May 2022

Effects of different exercise training programs on the functional performance in fibrosing interstitial lung diseases: a randomized trial

PONE-D-21-37890R1

Dear Dr. Shafiek,

We’re pleased to inform you that your manuscript has been judged scientifically suitable for publication and will be formally accepted for publication once it meets all outstanding technical requirements.

Kind regards,

Walid Kamal Abdelbasset, Ph.D.

Academic Editor

PLOS ONE

Additional Editor Comments (optional):

Reviewers' comments:

Reviewer's Responses to Questions

**Comments to the Author**

1. If the authors have adequately addressed your comments raised in a previous round of review and you feel that this manuscript is now acceptable for publication, you may indicate that here to bypass the “Comments to the Author” section, enter your conflict of interest statement in the “Confidential to Editor” section, and submit your "Accept" recommendation.

Reviewer #1: All comments have been addressed

Reviewer #2: All comments have been addressed

2. Is the manuscript technically sound, and do the data support the conclusions?

Reviewer #1: (No Response)

Reviewer #2: Yes

3. Has the statistical analysis been performed appropriately and rigorously? 

Reviewer #1: (No Response)

Reviewer #2: Yes

4. Have the authors made all data underlying the findings in their manuscript fully available?

Reviewer #1: (No Response)

Reviewer #2: Yes

5. Is the manuscript presented in an intelligible fashion and written in standard English?

Reviewer #1: (No Response)

Reviewer #2: Yes

6. Review Comments to the Author

Reviewer #1: (No Response)

Reviewer #2: All necessary corrections have been applied by the authors. The article has become acceptable. Well done.

7. PLOS authors have the option to publish the peer review history of their article (what does this mean?). If published, this will include your full peer review and any attached files.

Reviewer #1: No

Reviewer #2: **Yes: **Esedullah AKARAS

---

## [Editor Report · Acceptance letter]

18 May 2022

PONE-D-21-37890R1 

Effects of different exercise training programs on the functional performance in fibrosing interstitial lung diseases: a randomized trial 

Dear Dr. Shafiek:

I'm pleased to inform you that your manuscript has been deemed suitable for publication in PLOS ONE. Congratulations! Your manuscript is now with our production department. 

Kind regards, 

on behalf of

Dr. Walid Kamal Abdelbasset 

Academic Editor

PLOS ONE